# Adverse drug effects among students following mass de-worming exercise involving administration of Praziquantel and Albendazole in KEEA Municipality, Ghana

**Wisdom Akrasi[1], Augustine Suurinobah Brah[2], Mainprice Akuoko Essuman[1], Viona Osei[1], Alex Boye[1] ***

**1** Department of Medical Laboratory Science, School of Allied Health Sciences, College of Health and Allied Sciences, University of Cape Coast, Cape Coast, Ghana, **2** Department of Biomedical Sciences, School of Allied Health Sciences, College of Health and Allied Sciences, University of Cape Coast, Cape Coast, Ghana

* aboye@ucc.edu.gh

## Abstract

### Background

To manage the deleterious effects of parasitic infections such as lymphatic filariasis (LF) and schistosomiasis among school children, most countries including Ghana make use of mass drug administration (MDA). Although MDA has proven effective in reducing worm burden, unfortunately adverse drug effects (ADEs) post-MDA are derailing the gains and also remain poorly monitored. The study assessed incidence and factors associated with ADEs among students following a school-based mass de-worming exercise involving administration of Praziquantel (PZQT) and Albendazole (ADZ) against LF and SCH at Komenda-Edina-Eguafo-Abirem (KEEA) Municipal.

### Methodology

After fulfilling all ethical obligations, a total of 598 students aged 5–20 years who received PZQT or ADZ monotherapy or a combination of the two (PZQT + ADZ) as part of the mass de-worming exercise were recruited through quota and random sampling. Bodyweight and height of students were measured and body mass index (BMI) calculated. Students were orally interviewed to obtain information such as age, sex, intake of diet before taking drugs. Subsequently, students were monitored over 24 hours post-MDA for cases of ADEs. Descriptive statistics and logistic regression analysis using SPSS version 26 was used to describe data collected and to determine associations between incidence of ADEs and predictor variables.

### Principal findings

Out of the 598 students, 243 (40.64%) represented by 124 males (51.03%) and 119 females (48.97%) with mean (SD) age of 13.43 (2.74) years experienced one or more forms of ADE. In decreasing order, the detected ADEs included headache (64.6%), Abdominal pain

**Data Availability Statement:** Data are available from the University of Cape Coast Institutional Repository (ir.ucc.edu.gh).

**Funding:** The authors received no specific funding for this work.

**Competing interests:** The authors have declared that no competing interests exist.

(48.6%), fever (30.0%), diarrhea (21.4%) and itching (12.8%). Multivariable statistical analysis showed that age 5–9 years (OR: 2.01, p = 0.041) and underweight (OR: 2.02, p = 0.038) were associated with incidence of ADEs. Compared with students who received combination therapy, students who received ADZ only (OR: 0.05, p < 0.001) and PZQT only (OR: 0.26, p < 0.001) had low cases of ADEs. Gender and diet intake before MDA were not associated with ADE incidence.

## Conclusion

ADE incidence was common among students in the KEEA municipality. Age, underweight, and double dosing were associated with increase in ADE incidence, while gender and food intake were not associated with increase in ADE incidence. The Disease Control Unit of the Ghana Health Service should incorporate stringent ADE monitoring in post-MDA surveillance in the National MDA program in order to be able to detect, manage and report ADEs to inform planning for future MDA programs. Such initiatives will help not only in improving effectiveness of MDA programs but also identify high risk groups and exact strategies to reduce negative influence of ADE on MDA coverage and anthelminthic drug compliance.

## Author summary

Neglected tropical diseases such as lymphatic filariasis and schistosomiasis pose major health threat in poor tropical regions of the world. MDA is the preferred strategy mostly used in the treatment and management of lymphatic filariasis and schistosomiasis. Although MDA has proven effective since its adoption in 2000, however, most if not all post-MDA surveillance have focused on efficacy but not ADEs post-MDA. The present study assessed incidence of ADEs and factors that may contribute to ADE incidence among students treated with PZQT or ADZ monotherapy or combination of the two (PZQT + ADZ) as part of an annual District mass de-worming exercise against lymphatic *filariasis* and schistosomiasis at KEEA Municipality. Incidence of ADEs post-MDA, although mostly transient and manageable, was quite common among the study population. It is recommended that the Disease Control Unit of the Ghana Health Service should consider reviewing post-MDA surveillance to include ADE monitoring and management.

## Introduction

Helminth infections in humans are dominated by schistosomiasis, lymphatic filariasis and soil-transmitted helminthiasis and they afflict vulnerable populations although mortality is rare [1]. It is estimated that more than 200 million people are afflicted with schistosomiasis worldwide, with 85% of those cases in Sub-Saharan Africa [2,3]. Also, it is reported that over 1.2 billion people live in lymphatic filariasis endemic regions with approximately 25% suspected to be suffering from the infection [4,5]. Helminth infections have devastating health consequences for the infected populations, mostly children. For example, helminthic infections negatively affect physical and intellectual development of children. Also, helminth infections may affect population, particularly health bills of individuals and even quality of work [4]. Most affected countries use MDA programs to treat and manage helminthic infections among highly susceptible populations such as school-going children. Although helminthic

infections afflict all age groups, but children are more susceptible not only due to their developing immune system but also due to their natural tendency to play which mostly expose them to infective stages of helminths, particularly, soil transmitted helminths. Year in year out, the clock is ticking for the completion of the Sustainable Development Goal 3 (SDG 3), target 3.3 (which aims to sustainably end the epidemics of AIDS, tuberculosis, malaria and neglected tropical diseases [NTDs] and combat hepatitis, water-borne diseases and other communicable diseases by year 2030) [6,7], which has necessitated the need to increase both MDA and coverage of highly co-endemic communities.

In the year 2000, the World Health Organization (WHO) launched the Global Programme to Eliminate Lymphatic Filariasis (GPELF) and the core aim was to disrupt the transmission cycle of *W. bancrofti* and *Brugia spp*. through annual MDA to the most at-risk populations for a period of 5–6 years. Eighty-one countries were found to be endemic from year 2000. By the year 2012, 73 countries were endemic, with 120 million people infected, over 1.4 billion people at risk, and 40 million people affected by LF-related morbidity [8]. Among the first countries to implement MDA was Ghana. For instance, the Ghana Filariasis Elimination Programme was birthed in 2000 [6,8] and the annual MDA was strictly based on the WHO guidelines for countries that are co-endemic for lymphatic filariasis and other helminthic infections [8]. Central to the strategy was the recommendation for the use of ivermectin (IVM) and ADZ treatments to be undertaken every year for identified endemic populations with antigen prevalence above 1% [8]. In Ghana implementation of these guidelines have suffered many challenges that necessitated review.

Conventionally, schistosomiasis, lymphatic filariasis and onchocerciasis are treated by using ADZ, PZQT, IVM, and mebendazole [9]. MDA has proven useful in many countries that prioritize its use. In helminth-endemic countries, mass de-worming among school children is mostly used as a cost-effective strategy to enhance school attendance [10,11]. In a study to assess the impact of seven years of MDA against *Schistosoma haematobium* infection in Zanzibar, a significant decline in prevalence was observed from 3.9% in 2011 to 0.4% in 2020 among adults and a decline from 6.6% in 2012 to 1.2% in 2019 among school children [12]. Similarly, MDA compliance was attributed to decrease in the prevalence of lymphatic filariasis in Nepal [13], Egypt [14] and Tanzania [15]. In view of these gains, mass de-worming among school children remains a major treatment/management program to contain helminthic infections in many developing countries including Ghana.

Anthelminthic drugs used in MDA programs are without side effects. For instance, school going children sometimes suffer choking [3] and other ADEs such as headache, itching, abdominal pain, body weakness and dizziness from administration of ADZ and PZQT [16,17]. Fear for ADEs associated with MDA is considered a major factor that contribute to low MDA participation [6,18,19]. These drawbacks impact negatively on MDA effectiveness with respect to compliance with anthelminthic drug ingestion especially when school children or their guardians does the administration [20]. Advanced countries and some emerging economies that subscribe to MDA to manage helminthic infections do have post-MDA surveillance programs to monitor ADEs associated with MDA in order to effectively manage them to ensure patient safety [21]. Most developing countries in Sub-Saharan Africa including Ghana are commonly faced with inadequate expertise, financial and technical resources needed to undertake comprehensive MDA exercise to halt transmission of helminthic parasites in disease (schistosomiasis and lymphatic filariasis)-endemic communities. As a result, post-MDA surveillance in these countries are often limited to efficacy assessment of the anthelminthic drugs used in the MDA exercise [22]. An unintended consequence is that rare or common ADEs that are highly probable with the use of anthelminthic drugs during MDA exercise go unmonitored despite the fact that ADEs contribute to low MDA coverage [18,23,24]. Therefore, there

is the need to assess the nature and incidence of ADEs as well as factors that contribute to ADE incidence in disease-endemic communities.

This study assessed ADEs associated with mass administration of ADZ and PZQT either as monotherapy or combination therapy among school children in KEEA Municipality, Ghana. The present study has the potential to add to accumulating evidence that support the need for policy makers to review the current annual MDA program to include post-MDA ADE monitoring at least for the first 24 hours post-MDA.

## Methods

### Ethics statement

The study was approved by the Disease Control Unit of the Ghana Health Service who were undertaking MDA exercise at the schools and the Ghana Education Service, Central Region. The school authorities gave verbal consent for the study to be undertaken after having been informed of the purpose and importance of the study. All students were invited to participate after verbal informed consent from their parents and guardians had been obtained. Confidentiality, and anonymity were guaranteed during and after the study.

### Study design and population

Under the National Disease Control Programme, children in primary and junior high schools in endemic Districts and Municipalities receive oral administration of ADZ and PZQT as treatment against lymphatic filariasis and schistosomiasis annually. This study was conducted in November 2016 alongside the annual MDA exercise at the KEEA Municipality in an effort to monitor incidence of ADEs and possible factors that contribute to ADEs. Four public schools (Islamic junior high school and primary, Anglican junior high school and primary, Elmina M/A junior high school and primary, and Catholic Boys junior high school and primary) which were part of the MDA program were randomly selected. From the four schools, students aged 5–20 years in both Primary and junior high schools, permanently residing in communities in the KEEA Municipality with parent/guardian consent to participate were recruited into the study. Students with existing medical conditions or known allergy to the ADZ and PZQT were excluded from the study. The protocol for the drug administration program and the study were based on WHO guidelines for community-based implementation of schistosomiasis and soil transmitted helminths control programs [25].

### Sample size calculation

The study adopted quota and random sampling methods to select participants for the study. To undertake the quota sampling technique, the number of students in each school was obtained from school managements. The total population of students in the 4 selected schools, according to the school's records was 3,815. The sample size was calculated as described previously [26,27]. Briefly, a sample size of 498 was determined based on an estimate of incidence of ADEs of 39.2% from a previous study [28], a population size of 3,819, an acceptable margin of error of 4% and a confidence interval of 95%, using the StatCalc function of Epi Info software, Version 7.2.4.0 (Center for Diseases Control, Atlanta, Georgia, USA, and World Health Organization, Geneva, Switzerland). To make up for sampling errors, nonparticipation and missing information, a final sample size of 598 students were recruited for the study. Based on the relative number of students in each school, a proportion of participants was calculated for each of the schools as follows: Islamic junior high school and primary 128, Anglican junior high school and primary 160, Elmina M/A junior high school and primary 178, and Catholic

Boys junior high school and primary 132. The respective students were then randomly sampled from each of the schools for the study.

## Study area

This study was conducted in the KEEA municipal (Fig 1) in the Central Region of Ghana. KEEA municipality covers an area of 452 km$^2$ and is located between longitude 1° 20′ West and 1° 40′ West and latitude 5° 05′ North and 5° 15′ North. There are five sub-districts namely Kissi, Elmina, Komenda, Ankaful and Agona sub-districts. The municipality has an estimated

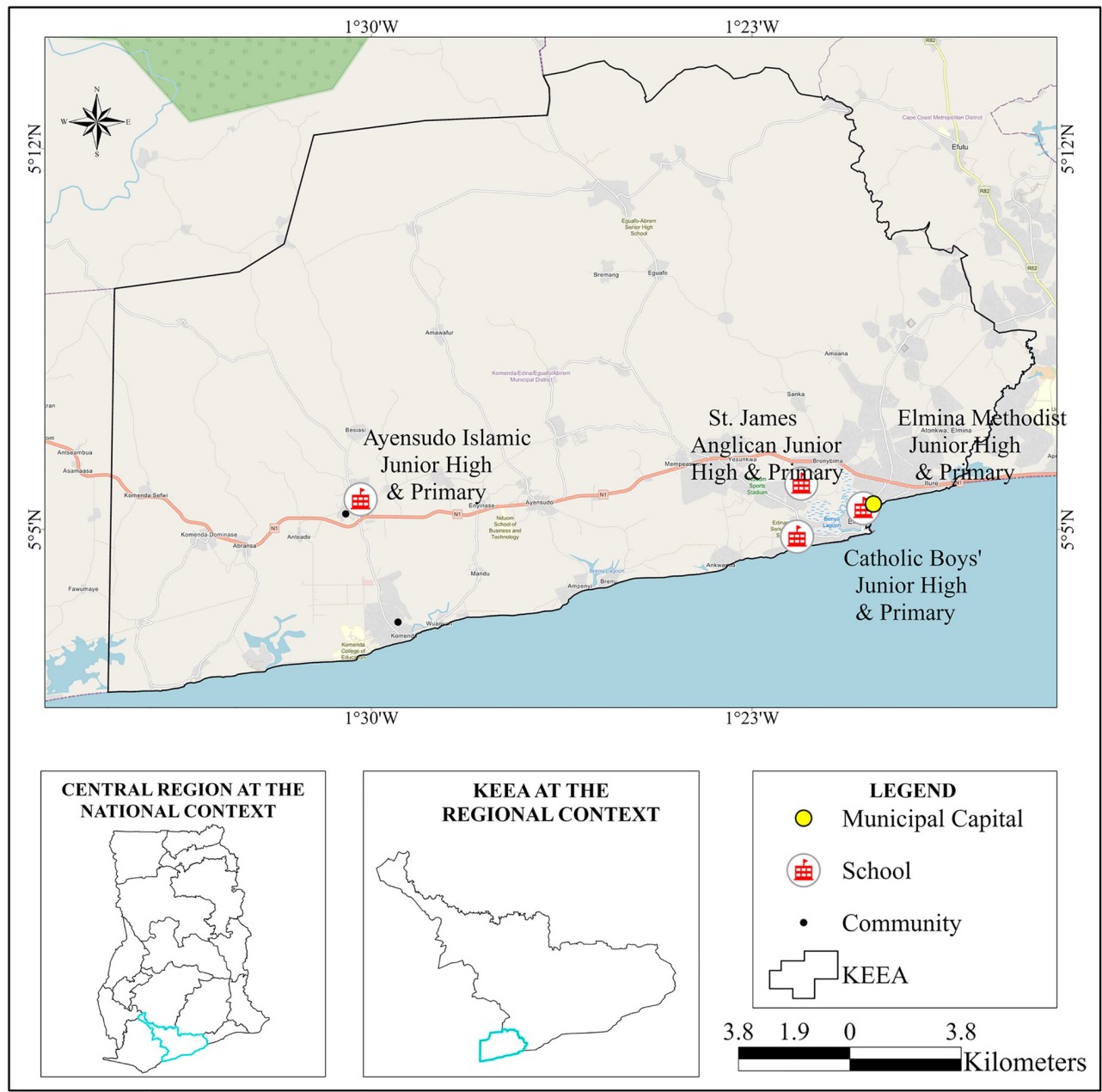

**Fig 1. Map of Komenda Edina Eguaafo Abirem (KEEA) Municipality.** (The map was created by the authors).

population of 144,705, with 69,665 (48.1%) being males and 75,040 (51.9%) females. It is estimated that 35,225 children in the municipality are in primary and junior high school. The majority of the people (70%) in the municipality reside in the rural areas with poor road and transport network. Poor access to safe water coupled with lack of proper sanitation, bathing and toilet facilities in this area contributes to a high prevalence of soil-transmitted helminthiasis especially in infants and pre-school children [29].

## Drug administration

Doses of ADZ and PZQT were administered according to WHO-dose pole format in which dosing is required to be based on height. Although the WHO-dose pole has been used for the past 20 years and validated in multiple populations, recent reports have questioned its accuracy especially when used in groups whose height and bodyweight do not correlate [30]. The present study used the updated format [30,31]. After measurement of bodyweight and height of students by health professionals from the Disease Preventive Unit, Ghana Health Service, Central Region, students were subsequently given doses of ADZ and PZQT (donated by Merck KGaA, Darmstadt, Germany) based on bodyweight and height. Weight and heights of students were measured prior to administration of drugs. Drugs were administered to students at their schools during first (10 am– 11 am in the morning) and second (1 pm– 2 pm) break periods. Each student was given one tablet of ADZ (400 mg). PZQT (40 mg/kg) dose was determined by using the height of students according to WHO-dose pole format [30] and the District Health Directorate's guidelines as follows: 94–109 cm received 1 tablet (40 mg/kg); 110–124 cm received 1½ tablets (60 mg/kg); 125–137 cm received 2 tablets (80 mg/kg); 138–149 cm received 2½ tablets (100 mg/kg); 150–159 cm received 3 tablets (120 mg/kg); 160–177 cm received 4 tablets (160 mg/kg) and $\geq$ 178 cm received 5 tablets (200 mg/kg). Some students were given a combination of both drugs. The original plan for the drug administration was to co-administer ADZ and PZQT against lymphatic filariasis and schistosomiasis one drug after the other 3 hours apart in line with WHO guidelines [25]. However, some students experienced immediate adverse effects after taking the first dose of ADZ. For such students, the second drug was not given. Co-administration of PZQT and ADZ has been reported to be efficacious and safe for the control of schistosomiasis and soil-transmitted helminthiasis in areas where both infections are endemic [25,32].

## Interview and measurements

Students were randomly selected after MDA and interviewed. Under our guidance and that of their teachers, students were made to answer few questions including: age, sex, intake of diet before taking drugs and whether they experienced any adverse effects such as headache, fever, cough, dizziness, diarrhea, abdominal pains, or vomiting/nausea within 24 hours of drug intake. Additionally, body weight and height were measured and body mass index (BMI) calculated. BMI of students were classified as follows: underweight ($< 18.5$ kg/m$^2$), normal weight (18.5–24.9 kg/m$^2$), overweight (25–29.9 kg/m$^2$), or obesity ($> 30$ kg/m$^2$) [33]. Any adverse events experienced by the students immediately after taking the drugs up to 24 hours after drug intake were observed and recorded by the researchers with the assistance of the health personnel involved in the administration of the drugs.

## Safety monitoring and ADEs management

Health personnel at health facilities in the study area were trained in handling possible adverse events as a result of drug administration. The documented self-reported and observed adverse effects of the two drugs were discussed during the training. A physician was available at the

district hospital to attend to emergencies. Fortunately, the immediate ADEs experienced by students were mild and transient, and did not require any medical intervention.

## Statistical analysis

Categorical variables were described as numbers and percentages (%) and continuous variables were described as the mean and standard deviation (SD). The Shapiro–Wilk test was used to verify normality. Differences between groups were assessed by a 2-sample t test for normally distributed continuous variables, the Mann–Whitney U test for non-normally distributed continuous variables, and the Chi-square or Fisher exact test for categorical variables. Bivariable analysis and Logistic regression models were used to examine the significant risk factors associated with ADEs. Data was first collected into Microsoft Excel and analysed using SPSS version 26.0 (Statistical Package for the Social Sciences, Chicago, IL USA) with GraphPad Prism 8 (GraphPad Software, San Diego, CA, USA) being used to generate figures. All tests were 2-tailed with p-values < 0.05 considered statistically significant in all analysis.

## Results

### General characteristics of study participants

A total of 598 students (mean age: 13.56 ± 2.57 years) who took part in the MDA program at the study area and satisfied the inclusion criteria were involved in the study. Out of this number, 306 (51.2%) were males, 231 (38.6%) were in primary school with 367 (61.4%) in junior high school. The average height, weight and BMI of students recruited into the study were 144 ± 16.55 cm, 44.88 ± 11.60 kg and 22.16 ± 7.21 Kg/m$^2$ respectively with females being significantly taller (145.60 ± 15.55 vs 142.47 ± 17.34, p = 0.021) and heavier (45.94 ± 11.23 vs 43.87 ± 11.86, p = 0.029) than males. Based on gender, no significant difference in terms of age, class and BMI was observed among the students. Majority 527/598 (88.1%) of the students ate before the MDA (Table 1).

### Drug administration

Administration of drugs was based on students' height in accordance with WHO dose-pole guidelines. Table 2 summarizes distribution of students with respect to treatment with ADZ and PZQT either as monotherapy or combination therapy. Almost half the total number of students 291/598 with slight female bias received combination therapy. Most of the students received PZQT monotherapy compared to ADZ monotherapy.

### ADE incidence

Of 598 students observed, 243 (40.64%; 124 males [51.03%]; 119 females [48.97%]; mean [SD] age, 13.43 [2.74] years) experienced one or more forms of ADE. The 243 students experienced 431 individual ADEs, an average of 1.8 ADEs per student. Of those who experienced ADEs, headache was the most reported 157 (64.6%) while itching 31 (12.8%) was the least reported among students (Fig 2A). None of the ADEs reported was fatal. Of the students who experienced ADEs, 172 (70.78%) complained of multiple ADEs with majority of students reporting abdominal pain and headache 44 (25.58%) but few reports for headache and itching 30 (17.44%) (Fig 2B).

### Factors that contribute to ADE incidence during MDA

A number of factors were identified to be significant risk factors for the incidence of ADEs among students when data was subjected to bivariate and logistic regression analysis. As

**Table 1. General characteristics of study participants.**

| Characteristics | Total (n = 598) | Male (n = 306) | Female (n = 292) | P-value |
|---|---|---|---|---|
| **Mean age (years)** | 13.56 ± 2.57 | 13.58 ± 2.63 | 13.54 ± 2.51 | 0.834 |
| **Age group n (%)** | | | | 0.609 |
| 5–9 | 41 (6.9) | 23 (56.1) | 18 (43.9) | |
| 10–14 | 300 (50.2) | 148 (49.3) | 152 (50.7) | |
| 15–20 | 257 (43.0) | 135 (52.5) | 122 (47.5) | |
| **Class** | | | | 0.480 |
| Primary | 231 (38.6) | 114 (49.4) | 117 (50.6) | |
| Junior high school | 367 (61.4) | 192 (52.3) | 175 (47.7) | |
| **Height (cm)** | 144.00 ± 16.55 | 142.47 ± 17.34 | 145.60 ± 15.55 | **0.021** |
| **Bodyweight (Kg)** | 44.88 ± 11.60 | 43.87 ± 11.86 | 45.94 ± 11.23 | **0.029** |
| **BMI (Kg/m$^2$)** | 22.16 ± 7.21 | 22.19 ± 7.45 | 22.13 ± 6.95 | 0.929 |
| **BMI groups n (%)** | | | | 0.126 |
| Underweight | 50 (8.4) | 23 (46.0) | 27 (54.0) | |
| Normal weight | 342 (57.2) | 173 (50.6) | 169 (49.4) | |
| Overweight | 72 (12.0) | 31 (43.1) | 41 (56.9) | |
| Obese | 134 (22.4) | 79 (59.0) | 55 (41.0) | |
| **Eaten prior to drug intake** | | | | 0.673 |
| Yes | 527 (88.1) | 268 (50.9) | 259 (49.1) | |
| No | 71 (11.9) | 38 (53.5) | 33 (46.5) | |

BMI—body mass index (calculated as weight in kilograms divided by height in meters squared).

summarized in Table 3, ADE incidence was significantly common among students aged 6–10 years 24/41 (58.5%), attending Islamic primary and junior high school 79/128 (61.7%), underweight 24/50 (48.0%) and those who were given a combination therapy 173/291 (59.5%), and 2.5 tablets of PZQT 27/77 (35.1%). In a multivariable logistic regression, risk of ADE remained significantly associated with students aged 6–10 year (OR: 2.01, p = 0.041); and highest among students attending Islamic junior high school (OR: 4.65, p < 0.001) or Anglican primary and junior high school (OR: 3.44, p < 0.001). The risk of ADE was 2.02 times likely to occur among students who were underweight compared to obese students. Students who received combination therapy (ADZ and PZQT) were more likely to experience an ADE than those who received monotherapy (either ADZ only or PZQT only). Food consumption before MDA was not associated with ADE incidence.

## Discussion

This study assessed the incidence of ADEs among students after mass administration of ADZ and PZQT either as monotherapy or combination therapy in randomly selected schools in the KEEA Municipal, Central Ghana. An ADE incidence of 40.6% (243/598) post-MDA was recorded, most of which were multiple and occurred within the first few hours after drug administration. Headache and abdominal pain were the most reported ADEs. Age, underweight and double dosing (administration of two or more drugs at the same time or combination therapy) were associated with ADE incidence, whiles eating before MDA and gender were not associated with ADE incidence. These observations from the present study perhaps indicate that ADE incidence is a common phenomenon experienced by students in the study area and could be the case for other MDA exercises where ADE monitoring does not form part of post-MDA surveillance. Comparatively, observations made from the present study somewhat

**Table 2. Drug administration for participants at the selected schools.**

| Drug dose | Schools | | | | Total (N = 598) |
|---|---|---|---|---|---|
| | Islamic school (N = 128) | Anglican school (N = 160) | Elmina M/A (N = 178) | Catholic boys (N = 132) | |
| **ADZ only** | | | | | |
| 1 tablet | 24 (18.8) | 9 (5.6) | 12 (6.7) | 23 (17.4) | 68 (11.4) |
| **PZQT only** | | | | | |
| 1 tablet | 0 (0.0) | 0 (0.0) | 6 (3.4) | 1 (0.8) | 7 (1.2) |
| 1.5 tablets | 0 (0.0) | 2 (1.3) | 13 (7.3) | 0 (0.0) | 15 (2.5) |
| 2 tablets | 3 (2.3) | 3 (1.9) | 25 (14.0) | 17 (12.9) | 48 (8.0) |
| 2.5 tablets | 5 (3.9) | 19 (11.9) | 33 (18.5) | 20 (15.2) | 77 (12.9) |
| 3 tablets | 4 (3.1) | 4 (2.5) | 20 (11.2) | 19 (14.4) | 47 (7.9) |
| 4 tablets | 5 (3.9) | 1 (0.6) | 20 (11.2) | 16 (12.1) | 42 (7.0) |
| 5 tablets | 0 (0.0) | 0 (0.0) | 3 (1.7) | 0 (0.0) | 3 (0.5) |
| **ADZ and PZQT Combination therapy** | | | | | |
| (1)(1) | 1 (0.8) | 9 (5.6) | 3 (1.7) | 0 (0.0) | 13 (2.2) |
| (1.5)(1) | 2 (1.6) | 16 (10.0) | 4 (2.2) | 0 (0.0) | 22 (3.7) |
| (2)(1) | 10 (7.8) | 28 (17.5) | 9 (5.1) | 7 (5.3) | 54 (9.0) |
| (2.5)(1) | 12 (9.4) | 36 (22.5) | 21 (11.8) | 12 (9.1) | 81 (13.5) |
| (3)(1) | 30 (23.4) | 15 (9.4) | 5 (2.8) | 12 (9.1) | 62 (10.4) |
| (4)(1) | 30 (23.4) | 15 (9.4) | 4 (2.2) | 5 (3.8) | 54 (9.0) |
| (5)(1) | 2 (1.6) | 3 (1.9) | 0 (0.0) | 0 (0.0) | 5 (0.8) |

(1) (1): (One tablet of PZQT) (One tablet of ADZ)

(1.5) (1): (One and a half tablet of PZQT) (One tablet of ADZ)

(2) (1): (two tablets of PZQT) (One tablet of ADZ)

(2.5) (1): (Two and a half tablets of PZQT) (One tablet of ADZ)

(3) (1): (Three tablets of PZQT) (One tablet of ADZ)

(3.5) (1): (Three and a half tablets of PZQT) (One tablet of ADZ)

(4) (1): (Four tablets of PZQT) (One tablet of ADZ)

(4.5) (1): (Four and a half tablets of PZQT) (One tablet of ADZ)

(5) (1): (Five tablets of PZQT) (One tablet of ADZ)

ADZ–Albendazole; PZQT–Praziquantel

corroborate other studies conducted in similar settings whiles at the same time appear to contrast with others.

For instance, the recorded ADE incidence of 40.6% and was found to be higher compared to ADE incidence reported from other countries with similar disease-endemicity and population characteristics. Example, in Kenya, 186 pupils from schools in Ndia community who were orally dosed with ADZ (400 mg) and PZQT (40 mg/kg) reported ADE incidence of 39.2% [28]. Similarly, two separate studies from Sri Lanka reported 7.5% and 12.6% (estimated from 160 and 2,319 dosed study subjects respectively) ADE incidences after study subjects were administered diethylcarbamazine and ADZ [34,35]. In Recife, a Brazilian community, an ADE incidence of 23.6% was reported [36]. Also, an ADE incidence of 25% was reported when *Wucheraria bancrofti* infected individuals were administered triple dose of IVM, diethylcarbamazine and ADZ [37]. In India, a triple (IVM, diethylcarbamazine and ADZ) and double (diethylcarbamazine and ADZ) dosed study subjects (4,758 and 4,160 respectively) reported 8.3% and 6.4% ADE incidences [38]. In Papua New Guinea, triple dosed (IVM, diethylcarbamazine and ADZ) study subjects reported ADE incidence of 20% whiles those who received double (diethylcarbamazine and ADZ) dose reported an ADE incidence of 18% [39].

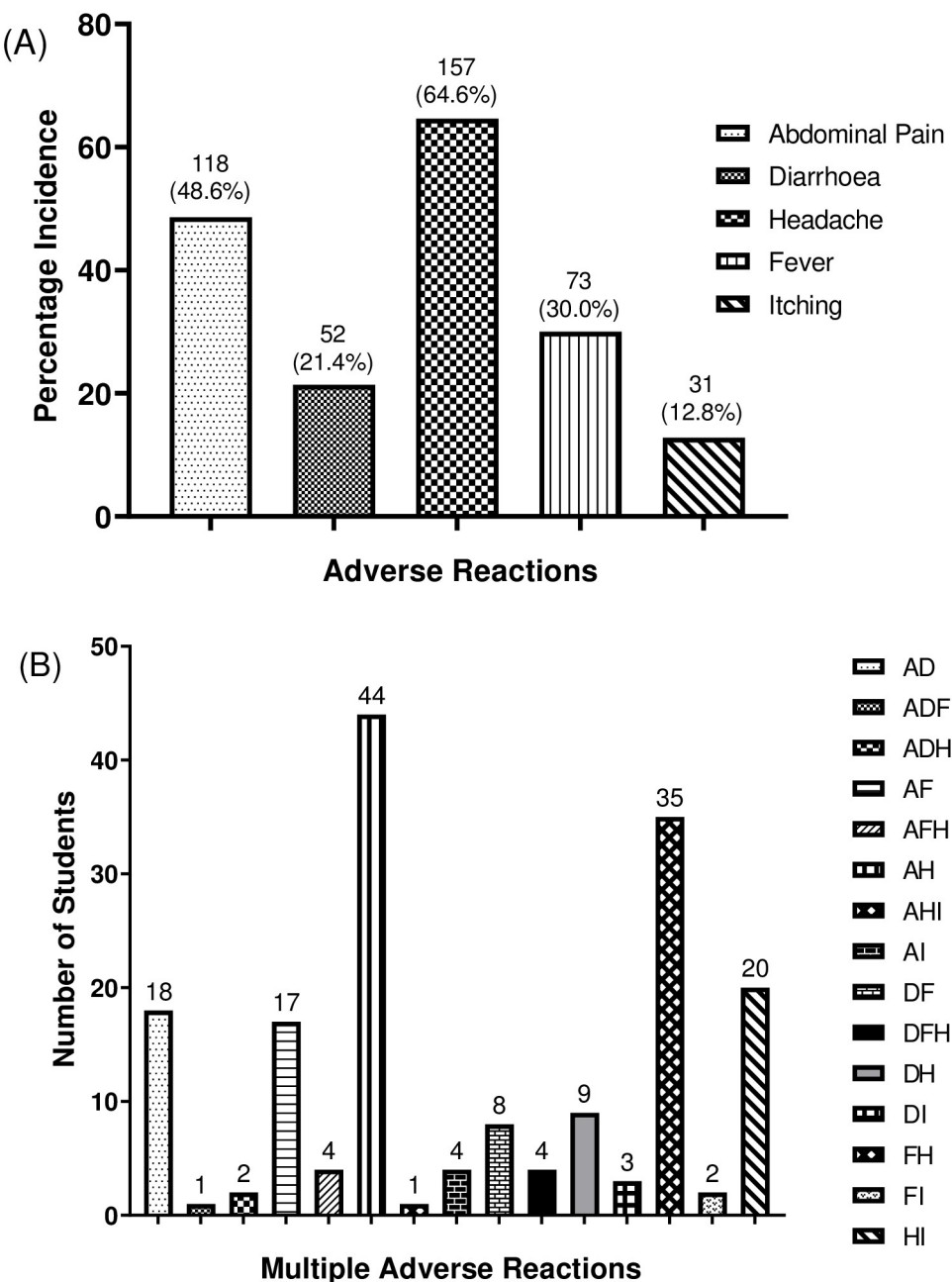

**Fig 2. ADEs reported by the students.** (A) kinds of ADEs commonly reported by students. (B) Multiple post-treatment ADEs reported by students. A—abdominal pains; D—diarrhea; F—fever; H—headache; I–itching.

In another development the estimated ADE incidence was relatively lower compared to ADE incidence of 83% recorded among students from two Kenyan communities [40,41]. Also, a study that systematically reviewed 55 studies on ADEs post-MDA reported a median ADE incidence > 60% among microfilariae infected individuals [42]. Clearly, there exist variation in ADE incidence even from studies conducted in communities with similar disease and population characteristics. Such disparities in ADE incidence from one study to the other is generally expected given the high heterogeneity in study characteristics such as sample size,

**Table 3. Relationship between Students'-related factors and the risk of adverse drug events based on bivariable and multivariable logistic analysis.**

| Variable | Total | No. (%) with ADEs | Chi square p-value | OR (95% CI) | P-Value |
|---|---|---|---|---|---|
| **Age Category** | | | **0.037** | | |
| 5–9 | 41 | 24 (58.5) | | 2.01 (1.03–3.93) | **0.041** |
| 10–14 | 300 | 113 (37.7) | | 0.86 (0.61–1.21) | 0.389 |
| 15–20 | 257 | 106 (41.2) | | 1 | |
| **Gender** | | | 0.954 | | |
| Male | 306 | 124 (40.5) | | 0.99 (0.72–1.37) | 0.954 |
| Female | 292 | 119 (40.8) | | 1 | |
| **School** | | | **<0.001** | | |
| Islamic primary and JHS | 128 | 79 (61.7) | | 4.65 (2.74–7.88) | **<0.001** |
| Anglican primary and | 160 | 87 (54.4) | | 3.44 (2.09–5.66) | **<0.001** |
| Elmina M/A primary and JHS | 178 | 43 (24.2) | | 0.92 (0.55–1.54) | 0.747 |
| Catholic Boys Primary and JHS | 132 | 34 (25.8) | | 1 | |
| **Class** | | | 0.164 | | |
| Primary | 231 | 102 (44.2) | | 1.27 (0.91–1.77) | 0.165 |
| JHS | 367 | 141 (38.4) | | 1 | |
| **BMI Classification** | | | **0.012** | | |
| Underweight | 50 | 24 (48.0) | | 2.02 (1.04–3.93) | **0.038** |
| Normal weight | 342 | 154 (45.0) | | 1.79 (1.18–2.74) | **0.007** |
| Overweight | 72 | 23 (31.9) | | 1.03 (0.56–1.90) | 0.929 |
| Obese | 134 | 42 (31.3) | | 1 | |
| **Drug Intake** | | | **<0.001** | | |
| **ADZ only** | 68 | 5 (7.4) | | 0.05 (0.02–0.14) | **<0.001** |
| **PZQT only** | 239 | 65 (27.2) | | 0.26 (0.18–0.37) | **<0.001** |
| **ADZ and PZQT combination therapy** | 291 | 173 (59.5) | | 1 | |
| **ADZ Only** | | | | | |
| 1 tablet | 68 | 5 (7.4) | | | |
| **PZQT Only** | | | **0.009** | | |
| 1 tablet | 7 | 0 (0.0) | | - | - |
| 1.5 tablets | 15 | 4 (26.7) | | - | - |
| 2 tablets | 48 | 14 (29.2) | | - | - |
| 2.5 tablets | 77 | 27 (35.1) | | - | - |
| 3 tablets | 47 | 11 (23.4) | | - | - |
| 4 tablets | 42 | 6 (14.3) | | - | - |
| 5 tablets | 3 | 3 (100) | | - | - |
| **ADZ and PZQT combination therapy** | | | 0.237 | | |
| (1)(1) | 13 | 6 (46.2) | | 1 | |
| (1.5)(1) | 22 | 11 (50.0) | | 1.17 (0.30–4.61) | 0.826 |
| (2)(1) | 54 | 28 (51.9) | | 1.26 (0.37–4.23) | 0.713 |
| (2.5)(1) | 81 | 45 (55.6) | | 1.46 (0.45–4.72) | 0.529 |
| (3)(1) | 62 | 42 (67.7) | | 2.45 (0.73–8.25) | 0.148 |
| (4)(1) | 54 | 38 (70.4) | | 2.77 (0.80–9.55) | 0.106 |
| (5)(1) | 5 | 3 (60.0) | | 1.75 (0.22–14.22) | 0.601 |
| **Eaten prior to drug intake** | | | 0.113 | | |
| No | 71 | 35 (49.3) | | 1 | |

*(Continued)*

**Table 3.** (Continued)

| Variable | Total | No. (%) with ADEs | Chi square p-value | OR (95% CI) | P-Value |
|---|---|---|---|---|---|
| Yes | 527 | 208 (39.5) | | 0.67 (0.41–1.10) | 0.115 |

- = not calculable. ADEs–Adverse drug effects; BMI—body mass index (calculated as weight in kilograms divided by height in meters squared); OR—odds ratio; CI–Confidence interval; JHS—Junior high school; ADZ–Albendazole; PZQT—Praziquantel

(1) (1): (One tablet of PZQT) (One tablet of ADZ)

(1.5) (1): (One and a half tablet of PZQT) (One tablet of ADZ)

(2) (1): (Two tablets of PZQT) (One tablet of ADZ)

(2.5) (1): (Two and a half tablets of PZQT) (One tablet of ADZ)

(3) (1): (Three tablets of PZQT) (One tablet of ADZ)

(3.5) (1): (Three and a half tablets of PZQT) (One tablet of ADZ)

(4) (1): (Four tablets of PZQT) (One tablet of ADZ)

(4.5) (1): (Four and a half tablets of PZQT) (One tablet of ADZ)

(5) (1): (Five tablets of PZQT) (One tablet of ADZ)

sampling method, type of anthelminthic drug, anthelminthic drug combinations (double or triple therapy), anthelminthic dose, expertise level of the investigators, intensity of infection, infection stage, timing of post-MDA surveillance as well as other unspecified host factors. For instance, whiles ADE incidence was common in 5–9 year groups in the present study, a similar study among students identified 7–16 year groups as having more immediate and delayed ADEs when they were dosed orally with ADZ and PZQT [43].

Increase in MDA coverage is one of the main objectives of every succesful MDA excercise in lymphatic filariasis and schistosomiasis co-endemic communities [44]. However, increase in MDA coverage in disease-endemic communities has not reflected in a proportionate increase in anthelminthic drug compliance. This gap between MDA coverage and anthelminthic drug compliance has been attributed largely to negative perceptions about ADEs associated with MDAs, particularly fear of ADEs [44–47]. Therefore, monitoring of ADEs post-MDA and proper eductation of community members on common ADEs associated with MDA prior to MDA exercise may improve anthelminthic drug compliance for effective treatment/management of lymphatic filariasis and schistosomiasis. In the present study, headache, abdominal pain, itching, fever and vomiting were the main ADEs reported by students who received ADZ and PZQT simultaneously. It is interesting to note that the observed ADEs were not much different from those reported in previous studies across disease-endemic countries. For instance, in 2011, Anto and colleagues had reported in a study from northern Ghana that headache was the most reported ADE when study subjects ingested a triple dose of ADZ, PZQT and IVM [48], indicating that mild ADEs such as headache may be commonly associated with multiple anthelminthic drug therapy. The spectrum of ADEs observed in the present study mirrors that of a previous study which reported abdominal pain, headache, itching, fever, vomiting, nausea and diarrhea when patients suffering from hydatid infection were administered with ADZ and PZQT [49], perhaps indicating that these spectrum of ADEs may be associated with PZQT and ADZ co-administration. The spectrum of ADEs was high among students who ingested PZQT only than those who ingested ADZ only, and this observation agrees with a number of reports. Example, Jaoko and colleagues had reported abdominal pain, headache, nausea, dizziness and fever in a study in Southeastern Kenya involving school pupils who ingested PZQT [43]. Similarly, Berhe and colleagues had reported abdominal cramps, bloody diarrhoea, dizziness and vomiting in a study in Ethiopia, when school children were given oral PZQT [50]. In Haiti, headache, dizziness and abdominal pain were the 3

most reported ADEs whiles in India, fever, headache and dizziness were the commonly reported ADEs following ingestion of PZQT [38,51]. Also, a report from Kenya indicate that ADEs are common with PZQT ingestion than ADZ ingestion [28]. Further, ADE bias to PZQT ingestion as observed in the present study corroborates observations made by Olds and colleagues who had reported that ingestion of PZQT by school children produced abdominal pain and bloody diarrhoea few hours after administration [41]. As a result, Olds and colleagues speculated that those ADEs could be directly linked with PZQT due to its bioavailability [41]. This assertion had been supported by an earlier report showing that ingestion of crushed PZQT tablets (40 mg/Kg) by pre-school children produced moderately severe ADEs, mostly body and face inflammation [52], suggesting that crushing of PZQT tablets perhaps increased its absorption and bioavailability. Most students reported two or more ADEs and this observation agrees with an earlier report that showed that most individuals reported multiple ADEs during post-MDA survellance [40] and this observation is expected because of systemic exposure of anthelminthic drugs which may exert many off-target effects. It should be pointed out that ADEs reported in the present study were general in nature unlike that of a study which reported that ADEs were limited to the gastrointestinal system when students ingested PZQT and ADZ [53]. Clearly, ADEs reported in the present study and that of other studies in Africa indicate that ADEs are treatment-related, drug-related, dose-dependent and also common with all MDA exercises irrespective of the geographical location of the communities, suggesting that health policy makers should review MDAs to include pre-MDA education of communities on ADEs as well as tailored treatment for ADEs during and after MDA execercises.

Age, underweight, and double dosing were associated with ADE incidence while gender and food intake were not associated with increase in ADE incidence. It was observed that students aged between 5–9 years had more ADEs compared to other age groups suggesting that perhaps younger students are more likely to experience ADE, an observation which contrasts with an earlier report which indicated that ADE incidence increases with age [54]. In view of this, perhaps more studies should be done to ascertain the relationship between age of children and ADE incidence post-MDA. Gender was not associated with ADE incidence in sharp contrast with a study which reported that gender differences were related to ADE incidence [54]. It is established that intake of food before or concurrently with drugs may have a significant effect on the pharmacokinetics of the drugs particularly their absorption, distribution, bioavailability, metabolism and elimination [55]. For instance, bioavailability of ADZ and PZQT is shown to be influenced by intake of high-fat foods prior to drug administration [56] which means that the net effect of these two drugs which depends on their bioavailability could be influenced by food intake. Students who ate before oral administration of ADZ and PZQT reported comparatively a lower ADE incidence relative to those who did not eat before drug administration (Table 3). It is possible that poor absorption due to lack of food in the stomach as the case may be among those students who did not eat prior to drug administration contributed to more ADEs that were related to gastrointestinal tract including abdominal pain and vomiting. Although there was a difference with respect to ADE incidence for those students who ate and those who did not before drug administration, however, this difference was statistically insignificant. Nonetheless, as long as those who ate recorded a lower ADE incidence suggest that perhaps food intake may decrease ADE incidence among students and therefore food intake may be considered as a mandatory pre-condition for anthelminthic drug administration during MDA exercise.

## Limitations of the study

The findings of the present study should be considered in the light of these limitations. First, the study focused on ADEs 24 hours post-MDA, therefore ADEs that occurred beyond 24

hours post-MDA could not be tracked, which could be a possible source of underestimation of the ADE incidence. Secondly, despite attempts to standardize the coding and recording of ADEs, some side effects may have been misclassified or gone unnoticed, especially in the occasional chaotic settings in which children were de-wormed. Thirdly, selection of study sites was determined by drug distribution schedules, and awareness of the routine of ongoing de-worming activity at the municipality, therefore, it is unclear the degree to which our findings are representative of other sites, even within KEEA municipality. Lastly, sample size for the study may be inadequate to assess factors for certain ADEs, and there may be confounding variables that were not accurately removed. Nonetheless the present study has shown that ADEs are associated with PZQT and ADZ use in mass treatment of lymphatic filariasis and schistosomiasis in the study area which needs to be addressed.

## Conclusion

ADE incidence post-MDA was high among students in the KEEA Municipality. Age, underweight, and double dosing were associated with ADE incidence, while gender and food intake were not associated with increase in ADE incidence. Given that fear for ADEs associated with MDA is considered a major factor that contribute to low MDA coverage, it is suggested that the Disease Control Unit of the Ghana Health Service incorporate measures including feeding of students prior to MDA, ensure ADE monitoring post-MDA for the first 72 hours, manage and report ADEs among high risk groups. More importantly monitoring of ADEs post-MDA should be done in time blocks (preferably 6 hours intervals over three days post-MDA) in order to identify rare, common, immediate and delayed ADEs. Finally, health policy makers should review MDAs to include pre-MDA education of communities on ADEs as well as tailored treatment for ADEs that may arise during and after MDA exercercises.

## Acknowledgments

Our appreciation goes to the Ghana Health Service (Central Region), circuit supervisors of the MDA exercise, the head teachers, teachers and students of the schools included in the study.

## Author Contributions

**Conceptualization:** Alex Boye.

**Data curation:** Augustine Suurinobah Brah, Mainprice Akuoko Essuman, Viona Osei.

**Formal analysis:** Augustine Suurinobah Brah, Mainprice Akuoko Essuman.

**Funding acquisition:** Alex Boye.

**Investigation:** Wisdom Akrasi, Mainprice Akuoko Essuman, Viona Osei.

**Methodology:** Wisdom Akrasi, Alex Boye.

**Project administration:** Wisdom Akrasi, Alex Boye.

**Resources:** Wisdom Akrasi, Alex Boye.

**Software:** Augustine Suurinobah Brah, Mainprice Akuoko Essuman.

**Supervision:** Alex Boye.

**Validation:** Alex Boye.

**Visualization:** Augustine Suurinobah Brah, Mainprice Akuoko Essuman.

**Writing – original draft:** Wisdom Akrasi, Augustine Suurinobah Brah, Mainprice Akuoko Essuman, Viona Osei, Alex Boye.

**Writing – review & editing:** Augustine Suurinobah Brah, Mainprice Akuoko Essuman, Viona Osei, Alex Boye.

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
