## [Decision Letter · Decision Letter 0]

11 Sep 2021

Dear Dr Boye,

Thank you very much for submitting your manuscript "Adverse drug effects among school pupils following mass de-worming exercise involving administration of praziquantel and albendazole in KEEA Municipal, Ghana" for consideration at PLOS Neglected Tropical Diseases. As with all papers reviewed by the journal, your manuscript was reviewed by members of the editorial board and by several independent reviewers. They found manuscript interesting but raised various concerns in methodology (study instrument validation, sampling technique, analysis) and interpretation of results. Please also refer to the attached files.In light of the reviews (below this email), we would like to invite the resubmission of a significantly-revised version that takes into account the reviewers' comments. We cannot make any decision about publication until we have seen the revised manuscript and your response to the reviewers' comments. Your revised manuscript is also likely to be sent to reviewers for further evaluation.

Sincerely,

Tauqeer Hussain Mallhi, Ph.D

Associate Editor

Jennifer Keiser

Deputy Editor

Dear Authors, thank you for submitting in PLOS NTD. Your manuscript has been assessed by relevant experts from the field. They found manuscript interesting but raised various concerns in methodology (study instrument validation, sampling technique, analysis) and interpretation of results. It is requested to please consider the comments of reviewers.

Reviewer's Responses to Questions

**Key Review Criteria Required for Acceptance?**

**Methods**

-Are the objectives of the study clearly articulated with a clear testable hypothesis stated?

-Is the study design appropriate to address the stated objectives?

-Is the population clearly described and appropriate for the hypothesis being tested?

-Is the sample size sufficient to ensure adequate power to address the hypothesis being tested?

-Were correct statistical analysis used to support conclusions?

-Are there concerns about ethical or regulatory requirements being met?

Reviewer #1: Methods

Excellent methodology; however, authors should include the study protocol if there any use for the data collection, how the interview conducted and what are the questions asked? However, it is only a suggestion that authors include a supplementary file or an appendix. 

2.1. Ethical Consideration 

“The study was approved by the Diseases Control Unit of Ghana Health Service who were 106 undertaking the mass drug administration exercise at the schools and Ghana Education Service 107 (Central Region)”

Please mentioned ethical approval number with year 

All school pupils were invited to participate….please update people to students throughout the manuscript.

Reviewer #2: Somewhat

Reviewer #3: The points below are the main comments (but not limited to):

‎1. Lines 117-118 (Study area):‎ ‎The number of primary students ‎‎(25,299) is more than two times higher than pupils at ‎junior high school (9,926). However, in the present study, the proportion of Juniors is ‎higher (61.4%). So, the sample size is not representative for the population.‎

‎2. Lines 122-129 (Study Population):‎ The authors stated that the total study number taken was 598 school pupils, and they ‎were ‎randomly selected. Also, in the abstract authors reported using a multi-stage ‎stratified sampling technique. The design and sample strategy process were not clear, authors did not ‎mention how did they choose the schools from the 4 communities? How did they choose ‎the primary students ‎and junior students from each school and each community?‎

‎3. The authors did not provide any information about the sample size calculation?‎

‎4. It is not known when the study was conducted, no date? ‎

‎5. Since it is a mass-deworming activity (no diagnosis needed), the authors did not clarify why ‎some school pupils received a combination ‎of both drugs while others did not?‎

‎6. Line 123 (study population): age included 5-19 years. However, in table 3, there is a group for ‎ages 16-20 years.‎

Reviewer #4: Are the objectives of the study clearly articulated with a clear testable hypothesis stated?

Yes. 

Is the study design appropriate to address the stated objectives? Is the population clearly described and appropriate for the hypothesis being tested?

The study design and duration of the study are missing. how the four primary and junior high schools residing in the 4 communities were selected and how the 598 samples were proportionally allocated to these four primary and junior high schools is not clear. Clarification required on these statements. 

Is the sample size sufficient to ensure adequate power to address the hypothesis being tested?

Yes.

How the reported adverse drug events were classified as mild, moderate and severe is missing in the manuscript. Needs definition under the method section. 

Are there concerns about ethical or regulatory requirements being met? 

Yes. All requirements met.

Reviewer #5: • Your methods section although was almost complete, it was not organized and needs to be reorganized and completed to simplify the understanding of this section:

Start your methods section with the first paragraph which could be titled “Study design and population” where you indicate the design for the study and restating the purpose of the study. In this paragraph you will briefly indicate the site or location where the study was conducted, how the data were collected, your targeted population (by age and level in school), the medication that were administered, and how did you do the monitoring. This paragraph will help you introduce the reader to your following sections of the methods. Then, your “Ethical Consideration” section will be kept as is. The following two sections should be the “Study Area” and “Study Population”. The following sections should be “Drug Administration”, “Interview and Measurements”, “Safety Monitoring and Adverse Drug Events Management”, and “Statistical Analysis”.

In page 5, in the “Study Population” section: Reorganize this section to start with the justification why these studies were selected (currently the last sentence in the paragraph). Then, indicate the actual number of primary and junior high schools where data were collected before listing these schools’ names. Also, the number of students should not be mentioned in here as this should part of your results.

In page 5, in the “Drug Administration” section: I didn’t understand if all pupils should receive both medications or either one and why there would be discrepancy among the pupils in receiving the medication from the same MDA program. I saw that in the results as well, some pupil received both medications and others didn’t, you need to describe this in your methods. Also, this needs to be described in more details on how these medications were administered by the MDA program? Administered at home or in school, both tablets at the same time or a tablet on each day or ….? This would make a difference in the interpretation of your results. This should be adjusted then this adjustment should be reflected on the rest of the manuscript.

In page 5, in the “Interview and Measurements” section: you need to indicate when the actual data collection was conducted as it relates to the day of MDA in the school.

In page 5, your “Statistical Analysis” section needs major revision and reorganization. You may start with the descriptive data analysis, followed by the inferential statistics and try to make it more specific. Then you need to provide your sample size calculation. Then, the level of significance for the study can be listed. Finally, you may end up by the statistical software that were used in your analysis.

**Results**

-Does the analysis presented match the analysis plan?

-Are the results clearly and completely presented?

-Are the figures (Tables, Images) of sufficient quality for clarity?

Reviewer #1: Results: I do not have any comments for this section.

Reviewer #2: Yes

Reviewer #3: ‎7. Table 1:‎

A. In the combination therapy cells, (4tablets) (1 tablet), row number 6, the total number ‎‎of students who received it is 54, however, the number of males 28, and females 38 ‎giving a different number of 66.

 ‎

B. The percentages for all rows equal 100% but for (4tablets) (1 tablet), row number six, ‎for males 51.9% and females 61.3% (114.2%). Not to say that Chi-square test results ‎‎(percentages) are not adequately presented. More elaboration on this point next.‎

‎8. The name of tests performed in the headnotes or footnotes of tables are not available.‎

The major contradictions presented in the study:‎

‎1.‎ For tables 2, 3, 4, the percentages for chi-square results are presented incorrectly and ‎‎therefore there are major flaws in its presentation. To clarify this point please see table 4, ‎the ‎association between food intake and side effects. In the table, the total number for ‎those who eat ‎and experience side effects is “74 (72.5%)” and those who do not eat and ‎experienced side ‎effects is “28 (27.5%)”. However, going back to table 1, the total ‎number of those who do not ‎eat is 71 and those who do not eat and experienced side ‎effects is 28 (table 4) so 28/71 ‎equals 39%, meaning 39% of those who did not eat have ‎experienced side effects. On the other ‎hand, the total number of those who eat is 527 (table 1), ‎and those who eat and experience side ‎effects are only 74 (table 4), so that means that 74/527 ‎‎(14% ) of those who eat experience side effects. ‎This is a different and contradicting ‎conclusion to the authors' results’ presentation where they stated ‎in lines 210-211 that ‎‎“Majority (72.5%) 74/102 of those who reported adverse effects were pupils ‎who ate ‎prior to drug intake”. Also, in the abstract line 28-29 “it was quite common among 10 -‎‎14 ‎years age group and higher among pupils who ate before taking the drugs”.‎

‎2.‎ Also the same thing for age groups. line 28-29 “it was quite common among 10 -14 years ‎age ‎group”. However, when we do proper calculation we find that side effects were more ‎common ‎among the age category 5-9. Because the total number of students at this age as ‎shown in table 1 ‎is 41, the number of side effects among this age group as shown in table ‎‎2 is 24 so 24/41 equals ‎‎58%. On the other hand, the total number of students at 10-14 ‎years old in table 1 is 300. The total side effects for ‎the 10-15 age category among ‎primary school and junior high school in table 2 and 3 is 156. So ‎the percentage will be ‎‎156/300 equals 52%. It is worth mentioning that when authors reported ‎side effects ‎among primary and junior students they use different classifications of age and I ‎included ‎the students aged 15 years and their experience side effects. This means that if we ‎remove ‎students who experience side effects and were aged 15 to match 10-14 age category in table 1 we will get even a lower ‎percentage than 52%.‎‎‎

‎3.‎ Lines 24-25 “Over one-third (243/598) of pupils who received the drugs complained of ‎at least ‎one form of an adverse drug effect” and Lines 216-217 “Among 243 pupils ‎who experienced ‎adverse effects; majority reported headache (64.6%) while a few ‎reported itching (12.8%) ‎‎(Figure 2)”. According to the authors 243 pupils “complained of ‎at least one form of an adverse drug ‎effect”. However. In figure 2 A, 118 students ‎experience abdominal pain, 157 experience ‎headaches, 52 diarrhea, 73 increase in ‎temperature, 31 itchings. So the total number of students ‎who experience at least one ‎form of side effects is 431 (73% of all students), which is a totally different number. Not only this, but in ‎figure 2B also, ‎the total number of students who experience more than one side effect equal 261, ‎higher than 243 number.‎

‎4.‎ Similarly, another contradicting result. For example, in figure 2A those who ‎experience at least ‎diarrhea was 52 (they experience it alone or together with ‎other side effects). However, in figure 2B, for those who experience diarrhea + other ‎‎side effects, the total is 106 students. How come that the number of those who ‎‎experience at least diarrhea in figure 2A is lower than than the number of students ‎who experience ‎diarrhea+other side effects in figure 2B !!!!!

‎5.‎ Table 5: Drug-specific post-treatment adverse effects among pupils. 

In this table, the ‎total number ‎of adverse effects for all medications for both males and females is 178 ‎which is different from the total number reported in the results’ text, and different ‎from the figures, a ‎new number!!!‎

Reviewer #4: Are the results clearly and completely presented?

The information like, how many of the pupils develop mild, moderate and severe adverse drug events is missing and how long the adverse drug events lasted is not known. Any intervention measures to manage those undesired events are missing. Clarification is required on the above statements.

Reviewer #5: • The results section:

In page 6, line 160: the sentence needs revision “A total of 598 Basic school pupils were recruited into the MDA program” this cannot be true. If you included all patients that received the medication and they total only 598 pupils then this is not an MDA. You need to revise this statement.

In the results section, you don’t restate the numbers from the tables you rather need to focus on the important number only.

Whenever you conduct inferential statistics, you need to interpret the results from these tests and report it whenever its significant. Otherwise, don’t conduct the analysis that you can not interpret its meaning. This was true for the results of the “Praziquantel only (Tablet)” in table 1, if you don’t know the interpretation of this analysis then just remove the p-values from the table as this has minimal to no value in your study and it is not your objective.

In page 8, line 188-189: the use of “post drug administration” give the reader a false sense that there were pupils that had ADE before drug administration or that you have monitored a control group who didn’t receive medication.

The comparison between male and female was not the purpose of the study but you kept comparing between them with no much value for the reader.

**Conclusions**

-Are the conclusions supported by the data presented?

-Are the limitations of analysis clearly described?

-Do the authors discuss how these data can be helpful to advance our understanding of the topic under study?

-Is public health relevance addressed?

Reviewer #1: Yes, conclusion supported by the data presented

Reviewer #2: Yes

Reviewer #3: The conclusion and discussion was based on incorrect data presentation. So, need to be re-written.

Reviewer #4: The conclusion of the study should be supported by the results reported in the manuscript.

Reviewer #5: � Due to the flaws in the analyses and presentation of the results, the authors were not able to construct informative discussion and conclusion section for the reader.

The limitation section should address all the actual limitations in the study. The efficacy of the medication was not your objective so it is not a limitation in your study.

**Editorial and Data Presentation Modifications?**

Reviewer #1: (No Response)

Reviewer #2: (No Response)

Reviewer #3: (No Response)

Reviewer #4: Thank you for the opportunity to review this paper. The team are required to do a major revision.

Reviewer #5: (No Response)

**Summary and General Comments**

Reviewer #1: Thank you for giving me the opportunity to review this manuscript. This is a very well written and easy to follow manuscript. I really liked the study findings and how they are reported. I would like to congratulate the authors on conducting this study in one of our key school going students in Low and middle income countries (LMIC) such as Ghana and raising much-needed voice.

I have some suggestions and queries that will help improve the quality of the manuscript. I am sure you will not be disappointed by the end product once these changes have been made. 

Abstract: The abstract is a little bit confusing. Please clearly state what do you want your readers to know and what is the key message in your abstract. For example, what do you want Disease Control Unit of the Ghana Health Service institute monitoring ADRs? 

Also needs to be change, school pupils can be changed into school going students? 

Introduction: Is it appropriate to discuss the significance of the study? What is the study's rationale? Previous literature from Ghana and school going students from other countries, if such data is available. Please highlight in the introduction part.

Please utilise current references; for example, references 3, 5, and 7 were published in 1997 and should be updated. 

3. R. J. Stoltzfus, M. Albonico, J. M. Tielsch, H. M. Chwaya and L. Savioli, "School-based 335 deworming program yields small improvement in growth of zanzibari school children after one 336 year," The Journal of Nutrition, vol. 127, no. 11, pp. 2187-2193, 1997. 

5. P. K. Das, K. D. Ramaiah, D. J. Augustin and A. Kumar, "Towards elimination of lymphatic 342 filariasis in india," Trends in parasitology, vol. 17, no. 10, pp. 457-460, 2001.

7. N. De Silva, H. Guyatt and D. Bundy, "Morbidity and mortality due to ascaris-induced 346 intestinal obstruction," Transactions of the Royal Society of Tropical Medicine and Hygiene, vol. 347 91, no. 1, pp. 31-36, 1997. 

Methods

Excellent methodology; however, authors should include the study protocol if there any use for the data collection, how the interview conducted and what are the questions asked? However, it is only a suggestion that authors include a supplementary file or an appendix. 

2.1. Ethical Consideration 

“The study was approved by the Diseases Control Unit of Ghana Health Service who were 106 undertaking the mass drug administration exercise at the schools and Ghana Education Service 107 (Central Region)”

Please mentioned ethical approval number with year 

All school pupils were invited to participate….please update people to students throughout the manuscript. 

Results: I do not have any comments for this section. 

Discussion: 

Where is the study strength? as the limitations shows separately, make it as separate heading. 

References: Please review formatting of references - again particularly for consistency with Journal title. 

Please ensure that someone independent and highly proficient in written English has thoroughly checked/edited your revised submission.

Reviewer #2: PLOS neglected tropical disease, 

Reviewer comments 

The research by Wisdom Akrasi and colleagues is an interesting piece of work on adverse effects of the commonly used de-worming drugs considered to be safe for ages. 

The conclusion supports the objective of the paper.

I have some questions and suggestions that might help improve the publication.

Title: long, may you revise to make it concise

Introduction: 

well narrated

The authors in the last paragraph said “In Ghana and most African countries, adverse events following mass drug administration are rarely monitored and recorded.” This statement is expected to reflect the intention of the study. Given that the drugs are widely used for deworming for ages in one hand due to their established safety profile, why you carried the current study. Is there any published study that alerts the need for monitoring? Especially in African children? In your introduction i did not found a clear argument for such important questions to be answered. Please enrich your quest for the study. In line, please also check your statement at line 90 “major set-back in using these drug……” 

Methods

Ethical consent issued should be clarified further. 

Study area: you better narrate the type and prevalence of NTDs in the municipality preferably in this section, or other appropriate section. 

I did not find the study period. 

What is the study design? The sample estimation methods were not clear as well. How about the sampling technique? How and what type of randomization applied? Why multistage? How about the inclusion and exclusion criteria? For instance, Line 145 says “….immediately after taking the medication up to the following two days.” How about before taking the medications.

Line 150 you said “A study physician was available at the district hospital to attend to emergencies.” Why? How many emergency admissions encountered? What was the role of the study physician? Hoping this is not a planning phase, these and related questions are expected to be answered with your statement. Please clarify your purpose and the occurrences. For instance your statement at the conclusion section stated here can be complimented with such clarifications “Although these adverse effects were transient and mild, they should not be overlooked during future…..” 

Result: 

What does this statement mean “A total of 598 Basic school pupils were recruited into the MDA program” Are you the one who recruited the MDA targets? Were the 598 children the MDA targets or the samples for safety study? If the MDA recruits, where is your sampling?

You said those receiving 2.5 gm of PZQT had shown more side effects than the other doses administered. Why do you think this happened? I did not see major explanations. This should better be discussed well in your next submission. 

Discussion: 

 Line 267: You said “Even though the efficacy of both drugs against intestinal worms was not ascertained in this study, their adverse effects give a vivid evidence of the extent of their bioavailability” Since your stud is not a pharmacokinetic study, I did not agree with this statement. 

I believe, the limitation of the study has to be addressed beyond the efficacy assessment. Please consider all possible limitations of your work to enable future researches in the area.

Reviewer #3: Generally, results presentation, interpretation, and conclusion need to be re-done. The same thing for discussion as many of ‎the major contradicting results have been discussed inadequately. I give here one example in lines ‎‎236-237 (but not limited to). Authors reported that “the highest frequency was observed in the 10–14 age group similar ‎to reports from Ethiopia [23]”. However, in the present study age 5-9 has more side effects ‎rate than 10-14 years and that the incorrect data presentation and interpretation lead to incorrect conclusion and discussion‎.

Reviewer #4: In general, the problem statement is not well stated (the need to monitor and document these adverse effects). The method section should be revised rigorously: study design employed, when and for how long the study conducted, how the samples were allocated proportionally to the selected schools and appropriateness of the statistical analysis used to determine association between variables (e.g., incidence of adverse drug events and food intake). It is not clear how the authors handle the reported adverse effects, including any medication used to manage these adverse effects.

Reviewer #5: (No Response)

PLOS authors have the option to publish the peer review history of their article (what does this mean?). If published, this will include your full peer review and any attached files.

Reviewer #1: No

Reviewer #2: Yes: Getachew Alemkere

Reviewer #3: No

Reviewer #4: No

Reviewer #5: No
---

## [Decision Letter · Decision Letter 1]

20 Mar 2022

Dear Dr Boye,

Thank you very much for submitting your manuscript "Adverse Drug Events Among Students Following Mass De-Worming Exercise Involving Administration Of Praziquantel And Albendazole In KEEA Municipality, Ghana" for consideration at PLOS Neglected Tropical Diseases. As with all papers reviewed by the journal, your manuscript was reviewed by members of the editorial board and by several independent reviewers. In light of the reviews (below this email), we would like to invite the resubmission of a significantly-revised version that takes into account the reviewers' comments. 

We cannot make any decision about publication until we have seen the revised manuscript and your response to the reviewers' comments. Your revised manuscript is also likely to be sent to reviewers for further evaluation.

Sincerely,

Tauqeer Hussain Mallhi, Ph.D

Associate Editor

Jennifer Keiser

Deputy Editor

Reviewer's Responses to Questions

**Key Review Criteria Required for Acceptance?**

**Methods**

-Are the objectives of the study clearly articulated with a clear testable hypothesis stated?

-Is the study design appropriate to address the stated objectives?

-Is the population clearly described and appropriate for the hypothesis being tested?

-Is the sample size sufficient to ensure adequate power to address the hypothesis being tested?

-Were correct statistical analysis used to support conclusions?

-Are there concerns about ethical or regulatory requirements being met?

Reviewer #4: (No Response)

Reviewer #5: Very much improved in term of clarity and organization.

**Results**

-Does the analysis presented match the analysis plan?

-Are the results clearly and completely presented?

-Are the figures (Tables, Images) of sufficient quality for clarity?

Reviewer #4: (No Response)

Reviewer #5: Your results section is much improved and became more meaningful.

Note: avoid saying “students used for the study ….” You can say “students included in the study …” or “students participated in the study…”.

**Conclusions**

-Are the conclusions supported by the data presented?

-Are the limitations of analysis clearly described?

-Do the authors discuss how these data can be helpful to advance our understanding of the topic under study?

-Is public health relevance addressed?

Reviewer #4: (No Response)

Reviewer #5: The conclusion in the text is limited compared to the conclusion in the abstract.

The limitations is clearly described.

The implication of the findings and the public health relevance is not fully addressed and can be improved.

**Editorial and Data Presentation Modifications?**

Reviewer #4: (No Response)

Reviewer #5: Most of these general points were previously indicated but was not addressed by the authors:

In page 6, line 163: the word “Majority” and “Municipality” should be with small letter.

In page 7, lines 177-183: in the sentence “Each student was given one (1) tablet of albendazole …” we know that one is (1), you don’t have to restate it into parenthesis. Just keep “one” in this case and the same is true in the following sentences.

In page 7, lines 180-183: the word “Tablet” or “Tablets” should be with small letter. 

In page 6 lines 162-163: the words “primary”, “junior high”, and “school” does not have to be started with capital letters. Only the name of the school should be started with Capital letter. This was the case in the result section as well, so you need to adjust it everywhere in your manuscript.

In page 9, line 224: “Body Mass Index (BMI)” this should be body mass index (BMI) but since it was mentioned in the methods then you should use the abbreviation directly.

In page 15, line 305: “Neglected Tropical Diseases (NTDs)” was defined twice in the same paragraph.

In page 15, line 306: What do you mean by basic schools? I think schools should be fine.

The manuscript may benefit from a revision by language editing service to improve the writing and flow of the paragraphs.

In page 16, line 346: “increase in body temperature (fever)” you don’t have to define fever. Everyone knows what fever means.

**Summary and General Comments**

Reviewer #4: The comments were corrected thoroughly.

Thank you all in advance.

Reviewer #5: The paper explored the incidence and type of adverse drug effects or events (ADE) among school pupils in KEEA Municipality of Ghana after mass drug administration in an effort to fight lymphatic filariasis and schistosomiasis. The study presented significant findings about the incidence of ADE. Although the authors’ revision has improved the manuscript, I think this still need to be revised to make this manuscript good for readers.

PLOS authors have the option to publish the peer review history of their article (what does this mean?). If published, this will include your full peer review and any attached files.

Reviewer #4: No

Reviewer #5: No
---

## [Decision Letter · Decision Letter 2]

29 May 2022

Dear Dr Boye,

Thank you very much for submitting your manuscript "Adverse Drug Events Among Students Following Mass De-Worming Exercise Involving Administration Of Praziquantel And Albendazole In KEEA Municipality, Ghana" for consideration at PLOS Neglected Tropical Diseases. As with all papers reviewed by the journal, your manuscript was reviewed by members of the editorial board and by several independent reviewers. In light of the reviews (below this email), we would like to invite the resubmission of a significantly-revised version that takes into account the reviewers' comments. 

Dear authors, thank you for revising the draft. There are still some concerns raised by the reviewers in the manuscript. Please refer to the attached files too. The referees are suggesting to improve the writing as well as methodological shortcomings in this manuscript.

We cannot make any decision about publication until we have seen the revised manuscript and your response to the reviewers' comments. Your revised manuscript is also likely to be sent to reviewers for further evaluation.

Sincerely,

Tauqeer Hussain Mallhi, Ph.D

Associate Editor

Jennifer Keiser

Deputy Editor

Dear authors, thank you for revising the draft. There are still some concerns raised by the reviewers in the manuscript. Please refer to the attached files too. The referees are suggesting to improve the writing as well as methodological shortcomings in this manuscript.

Reviewer's Responses to Questions

**Key Review Criteria Required for Acceptance?**

**Methods**

-Are the objectives of the study clearly articulated with a clear testable hypothesis stated?

-Is the study design appropriate to address the stated objectives?

-Is the population clearly described and appropriate for the hypothesis being tested?

-Is the sample size sufficient to ensure adequate power to address the hypothesis being tested?

-Were correct statistical analysis used to support conclusions?

-Are there concerns about ethical or regulatory requirements being met?

Reviewer #5: (No Response)

Reviewer #6: (No Response)

Reviewer #7: No

Reviewer #8: The method section clearly stated the objectives, the study design was appropriate to address the stated objectives. The population clearly stated and appropriated and sample size sufficient to ensure adequate power to address the hypotheses . Correct statistical analysis were used and no concerns about ethical or regulatory requirements.

Reviewer #9: -Are the objectives of the study clearly articulated with a clear testable hypothesis stated? : Yes, the study objectives are clearly stated although I have my reservations about the 'testability' of the hypothesis. Just because an individual experiences any effect after a drug is administered, does not mean that the effect was caused by the drug, especially for several of the very non-specific signs and symptoms usually reported in pharmacovigilance studies like the ones reported by this study's authors. Hence the need for something known as 'causality assessments' (e.g. the WHO-UMC system) which is a whole different ballgame, but which can help to produce really good quality evidence. The other thing is I think what the authors refer to as adverse drug events are actually just side effects (these words are used interchangeably but are not the same)

-Is the study design appropriate to address the stated objectives?: I am not sure especially as the authors did not provide in depth details as to exactly how the students were monitored for 24 hours after the MDA and how the 'ADEs' were defined. This study was conducted in school aged children who would have needed to go home after school hours, so how exactly were they monitored and data collected? Were the ADEs self reported by the participants or directly observed by the data collectors? How were the ADES defined, was 'fever' objectively measured with a thermometer or self reported? the same thing for 'diarrhea', how many loose stools did participants need to report before they were classed as having 'diarrhea' etc. Were the students asked to report any symptoms they experienced or shown a list of expected symptoms and asked whether they experienced any on the list (which could be a source of bias) etc. All these details need to be added in the 'interviews and measurements' subsection of the manuscript

-Is the population clearly described and appropriate for the hypothesis being tested?: Yes.

-Is the sample size sufficient to ensure adequate power to address the hypothesis being tested? Yes, I think so. Although I'm not sure why the authors used 4% as their margin of error instead of 5%, and why they used 3819 in their sample size calculation when the total number of students in the selected schools was 3815.

-Were correct statistical analysis used to support conclusions?. Yes.

-Are there concerns about ethical or regulatory requirements being met?: No. Although, the authors should include the actual approval numbers in their manuscript.

**Results**

-Does the analysis presented match the analysis plan?

-Are the results clearly and completely presented?

-Are the figures (Tables, Images) of sufficient quality for clarity?

Reviewer #5: (No Response)

Reviewer #6: (No Response)

Reviewer #7: Yes

Reviewer #8: The results are clear and well presented, analysis presented matches the plan

Reviewer #9: Does the analysis presented match the analysis plan?: Yes

-Are the results clearly and completely presented?: Yes

-Are the figures (Tables, Images) of sufficient quality for clarity?: Yes.

**Conclusions**

-Are the conclusions supported by the data presented?

-Are the limitations of analysis clearly described?

-Do the authors discuss how these data can be helpful to advance our understanding of the topic under study?

-Is public health relevance addressed?

Reviewer #5: (No Response)

Reviewer #6: (No Response)

Reviewer #7: No

Reviewer #8: The conclusions support the data presented, limitations clearly described and authors discussed how these data can be helpful. Public health is relevance and has been addressed.

Reviewer #9: -Are the conclusions supported by the data presented?: Yes

-Are the limitations of analysis clearly described?: Yes.

-Do the authors discuss how these data can be helpful to advance our understanding of the topic under study?: Yes

-Is public health relevance addressed?: Yes

**Editorial and Data Presentation Modifications?**

Reviewer #5: (No Response)

Reviewer #6: (No Response)

Reviewer #7: No

Reviewer #8: A few grammar error and omission such as line 101," it should read not without side effects"

Reviewer #9: (No Response)

**Summary and General Comments**

Reviewer #5: (No Response)

Reviewer #6: This manuscript is titled " Adverse Drug Events Among Students Following Mass De-Worming Exercise Involving Administration Of Praziquantel And Albendazole In KEEA Municipality, Ghana" At KEEA Municipality, Ghana, the authors examined the ADEs associated with mass administration of ADZ and PZQT either as monotherapy or in combination therapy among school students. Results showed that Students in the KEEA Municipality's designated schools had an unusually high number of ADEs after their MDA. 

Overall, this is a well-written manuscript. However, the authors make any contribution to the research literature in this area of investigation. 

Specific comments/inquiries are below:

No additional comments at this stage

Reviewer #7: No

Reviewer #8: A well written manuscript, that addresses a relevant and important issue.

Reviewer #9: 1. Why is the abstract in the manuscript submission system different from the one currently attached to the body of the manuscript? Did the authors forget to update the abstract in the manuscript submission system after the initial review?

2. Minor English language editing may also be necessary for some portions of the manuscript E.g. Line 12: Use ‘Many’ instead of ‘Most’, Line 101: 'not' is missing, Tables 1 and 3: 'ate' instead of eaten and so on.

3. Please note that there is no need to refer again to your results in the discussion section. Therefore all references to tables 1 and 3 etc. in the discussion section can be taken out. Similarly, percentages and/ proportions already reported in the results section e.g. lines 307 and 317 can also be taken out. 

4. The paragraph starting from line 354 can be edited for clarity. How exactly does the pharmacokinetics of albendazole explain its ADE profile?

PLOS authors have the option to publish the peer review history of their article (what does this mean?). If published, this will include your full peer review and any attached files.

Reviewer #5: No

Reviewer #6: No

Reviewer #7: No

Reviewer #8: No

Reviewer #9: No
---

## [Decision Letter · Decision Letter 3]

20 Jul 2022

Dear Dr Boye,

We are pleased to inform you that your manuscript 'Adverse Drug Effects Among Students Following Mass De-Worming Exercise Involving Administration Of Praziquantel And Albendazole In KEEA Municipality, Ghana' has been provisionally accepted for publication in PLOS Neglected Tropical Diseases.

Best regards,

Tauqeer Hussain Mallhi, Ph.D

Academic Editor

Jennifer Keiser

Section Editor

Reviewer's Responses to Questions

**Key Review Criteria Required for Acceptance?**

**Methods**

-Are the objectives of the study clearly articulated with a clear testable hypothesis stated?

-Is the study design appropriate to address the stated objectives?

-Is the population clearly described and appropriate for the hypothesis being tested?

-Is the sample size sufficient to ensure adequate power to address the hypothesis being tested?

-Were correct statistical analysis used to support conclusions?

-Are there concerns about ethical or regulatory requirements being met?

Reviewer #5: (No Response)

Reviewer #8: The methods addresses the objectives, study design, population, sample size and the analysis appropriately

Reviewer #9: Please see my comments to the editor below

**Results**

-Does the analysis presented match the analysis plan?

-Are the results clearly and completely presented?

-Are the figures (Tables, Images) of sufficient quality for clarity?

Reviewer #5: (No Response)

Reviewer #8: The analysis presented matched the plan, results were clearly and completely presented and figures and tables were of sufficient quality and clarity

Reviewer #9: Please see my comments to the editor below

**Conclusions**

-Are the conclusions supported by the data presented?

-Are the limitations of analysis clearly described?

-Do the authors discuss how these data can be helpful to advance our understanding of the topic under study?

-Is public health relevance addressed?

Reviewer #5: (No Response)

Reviewer #8: The conclusions were supported by the data presented, limitations of analysis were clearly described and authors described how data can be helpful to advance understanding of the topic. Public health relevance was addressed.

Reviewer #9: Please see my comments to the editor below

**Editorial and Data Presentation Modifications?**

Reviewer #5: The paper now is much better than than the last three versions. However, this needs language revision as the writing of many part of this can be improved and benefit from language revision.

Reviewer #8: None

Reviewer #9: (No Response)

**Summary and General Comments**

Reviewer #5: (No Response)

Reviewer #8: This is a much improved manuscript, findings worth publishing

Reviewer #9: (No Response)

PLOS authors have the option to publish the peer review history of their article (what does this mean?). If published, this will include your full peer review and any attached files.

Reviewer #5: No

Reviewer #8: No

Reviewer #9: No

---

## [Editor Report · Acceptance letter]

8 Sep 2022

Dear Dr Boye,

We are delighted to inform you that your manuscript, " Adverse Drug Effects Among Students Following Mass De-Worming Exercise Involving Administration Of Praziquantel And Albendazole In KEEA Municipality, Ghana ," has been formally accepted for publication in PLOS Neglected Tropical Diseases.

Best regards,

Shaden Kamhawi

co-Editor-in-Chief

Paul Brindley

co-Editor-in-Chief
